

# Pick-up point recommendation strategy based on user incentive mechanism

Jing Zhang[1], Biao Li[1], Xiucai Ye[2] and Yi Chen[3]

[1] School of Computer Science and Mathematics, Fujian Provincial Key Laboratory of Big Data Mining and Applications, Fujian University of Technology, Fu Zhou, Fu Jian, China
[2] Department of Computer Science, University of Tsukuba, Ibaraki, Tsukuba, Japan
[3] Department of Computer and Information Security Management, Fujian Police College, Fu Zhou, Fu Jian, China

## ABSTRACT

In recent years, with the development of spatial crowdsourcing technology, online car-hailing, as a typical spatiotemporal crowdsourcing task application scenario, has attracted widespread attention. Existing researches on spatial crowdsourcing are mainly based on the coordinate positions of user and worker roles to achieve task allocation with the goal of maximum matching number or lowest cost. However, they ignores the problem of the selection of the pick-up point which needs to be solved in the actual scene of online car booking. This problem needs to take into account the four-dimensional coordinate positions of users, workers, pick-up point and destination. Based on this, this study designs a pick-up point recommendation strategy based on user incentive mechanism. Firstly, a new four-dimensional crowdsourcing model is established, which is closer to the practical application of crowdsourcing problem. Secondly, taking cost optimization as the index, a user incentive mechanism is designed to encourage users to walk to the appropriate pick-up point within a certain distance. Thirdly, a concept of forward rate is proposed to reduce the computation time. Some key factors, such as the maximum walking distance limit of users and task cost, are considered as the recommendation index for measuring the pick-up point. Then, an effective pick-up point recommendation strategy is designed based on this index. Experiments show that the strategy proposed in this article can achieve reasonable recommendation for pick-up points and improve the efficiency of drivers and reduce the total trip cost of orders to the greatest extent.

Corresponding author
Jing Zhang, jing165455@126.com

## INTRODUCTION

The concept of crowdsourcing was first proposed by Jeff Howe who defined it as "the practice of a company or organization to freely and voluntarily outsource work tasks that used to be performed by employees to a non-specific public network" (*Vander Schee, 2009*). Due to the rapid development of Internet technology and sharing economy model, new crowdsourcing models have gradually emerged and developed. To complete such crowdsourcing tasks, workers need to arrive at the designated place before the designated time (*Tong et al., 2017*). This new model, developed from traditional crowdsourcing

technology and integrating spatiotemporal information, is called spatiotemporal crowdsourcing, also known as spatial crowdsourcing. Spatial crowdsourcing uses a way of collecting group intelligence approach to complete tasks with spatiotemporal information, which has been widely promoted in recent years and its application fields are also constantly expanding. Currently, this concept generally refers to a new collaborative computing mode formed by assigning a task to a large number of mobile users with intelligent terminals. Because spatial crowdsourcing can complete large-scale complex tasks that are difficult for a single user to cope with alone through the collaboration of mobile users, such as data collection, logistics transportation, ride sharing, traffic information collection, environmental monitoring, *etc.* Therefore, it has wide application value and important research significance.

A typical spatial crowdsourcing system consists mainly of a system platform located in the cloud and many mobile users carrying smart terminals. The crowdsourcing service requester (User) will publish a spatial crowdsourcing task through the cloud system platform, which can usually be expressed as a series of geographically relevant data collection such as environmental detection, logistics transportation or ride-sharing tasks, *etc.* The platform then assigns these tasks to the right workers, or workers apply for crowdsourcing tasks that they are interested in. Subsequently, the worker moves to the designated location, performs and completes the task, then feeds back the results of the task to the initiator of the task through the intelligent terminal carried with him.

Figure 1 depicts a traditional spatial crowdsourcing scenario where users are matched to workers. Under the crowdsourcing model in the figure, there are two types of roles, the user $u_i$ and the worker $w_j$. The distance between any two roles is calculated by Euclidean distance. The platform matches the corresponding worker to the user based on the index of minimizing the movement distance of the worker. The figure shows the location and matching policies of four pairs of users and workers. After matching by the platform, the worker will move from the current location to the user's location to complete the crowdsourcing task. The blue arrow indicates the worker's movement path and distance. For example, the system matches the user $u_2$ with the nearest worker $w_2$, and the worker $w_2$ will move to the user's $u_2$ location to perform the task along the blue arrow mark.

However, the emergence of new crowdsourcing platforms has created new challenges for traditional crowdsourcing tasks. As a new kind of shared travel service, ride-hailing service is composed of drivers, users and platform. The platform match ride-hailing requests submitted by users with online drivers. As a practical application scenario of spatial crowdsourcing, the research on large-scale and real-time spatial crowdsourcing data processing technology is very hot. Pick-up point recommendation is an important part of ride-hailing service. After the task is assigned, the user and the driver need to go to the pick-up point respectively. In this scenario, three roles are involved: worker, user and work point.

Figure 2 depicts a spatial crowdsourcing task scenario involving three types of role positions. Under the crowdsourcing model in the figure, there are four pairs of users and workers and corresponding work points. The distance calculation standard in Fig. 2A adopts Euclidean distance calculation, and Fig. 2B adopts Manhattan distance calculation.

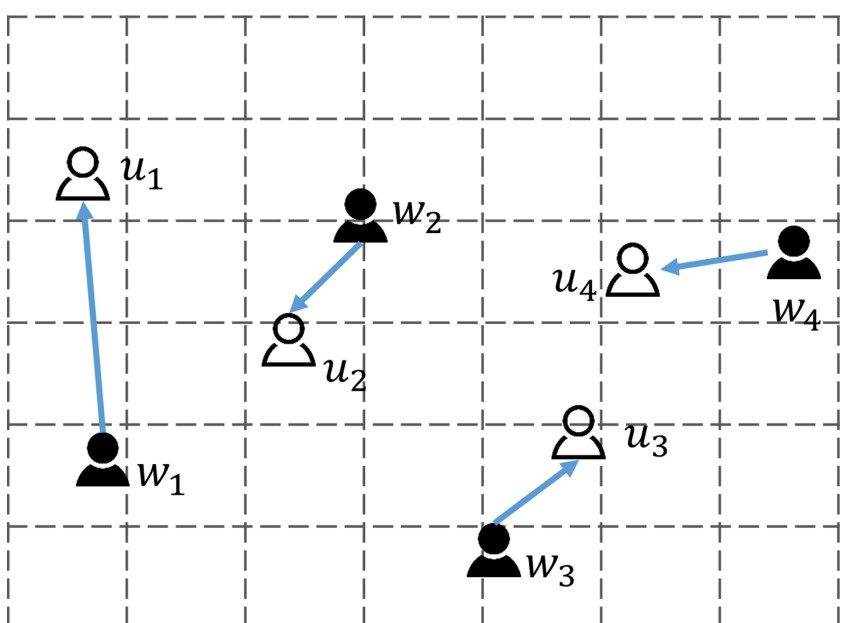

**Figure 1  Traditional crowdsourcing matching model.**

After being assigned by the platform, both the worker and the user move from their current location to the matching work point. The blue line is the distance moved by the worker, and the yellow line is the distance moved by the user. For example, in Fig. 2A, the platform matches the user $u_2$ and worker $w_2$ with the nearest work point $p_2$ , and the user $u_2$ and worker $w_2$ perform the task after moving to the work point $p_2$ along the yellow arrow and the blue arrow, respectively. The model in Fig. 2 extends the spatial crowdsourcing task to three dimensions. The question of selecting pick-up points can be attributed to the selection of work points in the spatial crowdsourcing task. The choice and arrangement of work points is crucial to the user experience. However, the important position of destination has been ignored in the practical problem of applying this to the ride-hailing service to recommend the pick-up points for users. It is assume that the platform has reached a match for the user and the driver. Furthermore, in order to recommend the appropriate ride point for the user, it is needed to considered that not only the location of the user and the driver, but also the destination factor.

Reasonable selection of ride points can reduce the detour distance of drivers, improve the efficiency of pickup, reduce user costs, and improve user travel experience. Therefore, the accurate and reasonable recommendation of the ride point has important research value. Based on this purpose and motivation, this study aims to study how to recommend suitable ride-hailing points for users in the spatial crowdsourcing environment with two-stage mobile tasks, so as to improve the efficiency of ride-hailing services.

Figure 3 depicts a four-dimensional space crowdsourcing task allocation and execution scenario that includes users, workers, work points and destinations, and calculates the mobile cost using the Manhattan distance. The figure shows two work points $p_1$ and $p_2$,

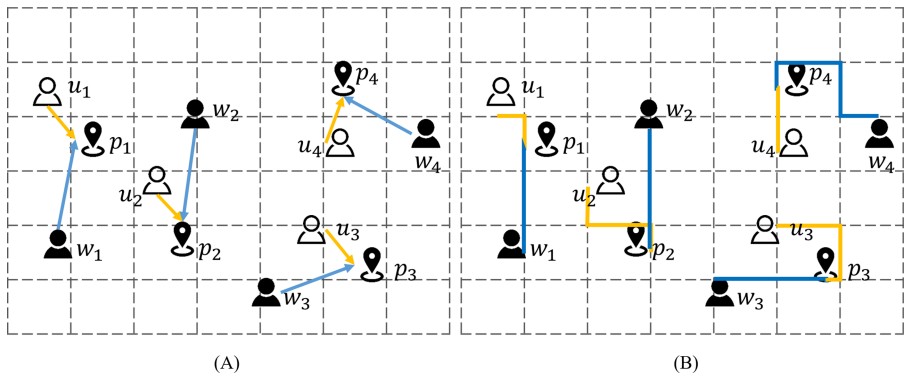

**Figure 2** 3D space crowdsourcing matching model.

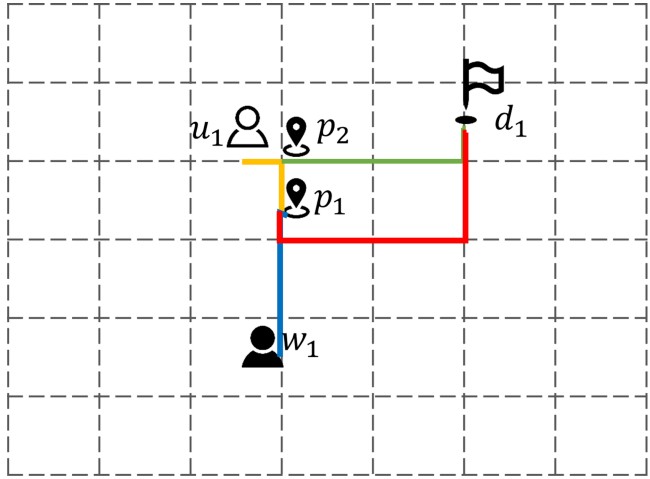

**Figure 3** Four dimensional space crowdsourcing work site selection.

both workers $w_1$ and users $u_1$ move from the current location to the designated work point, and then move together to the destination location $d_1$ after gathering. It can be seen from the figure that the selection of the work point has a significant impact on the subsequent moving distance. The red solid line is the subsequent moving track of the work point $p_1$, and the green realization is the subsequent moving track of the work point $p_2$.

However, in the real road network environment, the actual driving trajectory is more complicated due to the influence of driving restrictions and other factors. The actual distance cannot be calculated according to the ideal Manhattan distance in Figs. 2B or 3. The existing studies only consider the distance between the work point and the user and the worker. That is to minimize the moving distance between the two roles to the work point, while ignoring the actual situation that roles need to move to the destination point after reaching work point. Therefore, it is necessary to consider the four-dimensional space crowdsourcing scenario that includes four types of roles: user, worker, work point, and destination.

In summary, the main problem in the selection of work point in spatial crowdsourcing models is to recommend suitable pick-up points for users. However, most of the existing crowdsourcing matching models are three-dimensional spatial crowdsourcing matching models that only focus on the location of users, workers and work points. In this case, the influence of the pick-up point on the subsequent movement trajectory of the fourth dimension destination point is ignored, which may lead to the increase of the travel cost in the second stage. Secondly, the path distance calculation based on the European distance and Manhattan distance fails to take into account the one-way and two-way traffic restrictions and other factors in the real road network environment. When users are located at some complex traffic intersections, taking the above two kinds of distances standard as indicators may lead to inappropriate selection of pick-up points, resulting in the increase of the distance between the collection and the trip, and the overall travel cost is high. It also causes unnecessary energy waste, which is inconsistent with the existing energy-saving and emission reduction policies. Finally, if the algorithm based on ergodicity is used to selecting pick-up points, the calculation speed will be too slow, and the optimal solution cannot be obtained quickly in a large area of the real road network.

In response to the above challenges, this study designs a pick-up point recommendation strategy based on user incentive mechanism: Firstly, a new four-dimensional space crowdsourcing model is established, which uses real road network information and considers actual driving paths to be more realistic. Secondly, based on cost optimization as an indicator, user incentive mechanism is designed to encourage users to walk to the appropriate pick-up point within a certain distance. Thirdly, a concept of forward rate is proposed. Finally, key factors such as maximum walking distance limit and task cost are included in the recommendation index to measure the pick-up point, and an effective pick-up point recommendation strategy is designed. Experiments show that the proposed algorithm can achieve reasonable recommendation of pick-up points, improve the efficiency of drivers and reduce the cost of task execution. The specific contributions of this work are as follows:

(1) Aiming at the problem of pick-up point recommendation, a new four-dimensional space crowdsourcing model is proposed, including the spatial coordinate positions of four roles: user, worker, work point and destination point.

(2) The user incentive mechanism is designed. Using the difference between the execution cost of tasks from the current location and the recommended pick-up point location to the destination as user incentive value. Encourage users to walk within a certain distance to a suitable pick-up point within the maximum walking range to minimize travel costs.

(3) A pick-up point evaluation index based on the forward rate is proposed, which greatly reduces the calculation time of the pick-up point selection problem. On this basis, the optimal pick-up point is determined according to the incentive income ratio and other indicators.

(4) The pick-up point recommendation strategy is designed, and experiments are carried out on real data sets to prove the effectiveness of the pick-up point recommendation strategy proposed in this article.

The remainder of this article is organized as follows. The first section reviews the related work. The second section describes the problem. The third section proposes a protocol to solve the problem. The fourth section analyzes the experiments and results. At last, the current work is summarized and and the future work is prospected.

## RELATED WORK

At present, most spatial crowdsourcing studies mainly consider two types of objects, task and worker. The goal is to assign tasks through one-to-one or one-to-many matching mode under the premise of satisfying the constraints of task and worker (*Song et al., 2020*; *Li et al., 2020b*; *Liu & Xu, 2020*; *Gummidi, Xie & Pedersen, 2019*; *Pan et al., 2019*; *Wang et al., 2017*; *Tong et al., 2017*; *Yang, Tang & Zhang, 2019*). *Cheng et al. (2020)* propose to realize dynamic task allocation by coordinating worker resources among different platforms to maximize platform total revenue. *Chen et al. (2019)* study how to reduce the maximum waiting time in allocation to improve user satisfaction. *Qian et al. (2019)* propose an adaptive algorithm combining multi-arm gambling machines and batch processing to minimize the average waiting time of tasks. *Nei et al. (2020)* propose an online task allocation algorithm with comprehensive tripartite benefits. However, these studies not consider the effect of the selection of assigned work points on task execution.

*Seng et al. (2023)* research on the shared travel mode under crowdsourcing technology in smart cities. The directionality and complexity of urban roads and the distribution strategy of online ride-hailing platforms provide objective conditions for the existence of pick-up point recommendation. *Xie et al. (2023)* study the application of graph autoencoder based on regularization in recommendation algorithm. In the context of pick-up point selection, the conventional approach involves transmitting the user's current location to the server, subsequent to which the platform sets the user's positioning point as the pick-up point. However, in practical situations, due to the bidirectional nature of urban roads and the speed of drivers' movement, taking the user's location point as the pick-up point may increase the driver's driving cost, affect the driver's travel time, and thus indirectly affect the user's waiting time. Therefore, the spatial crowdsourcing task challenge of the online car-hailing platform has evolved task allocation from the location based on the two positions of user and driver to the based on the three positions of user, driver and pick-up point. In this way, the problem of spatial crowdsourcing for the three roles of users, workers and work points is addressed.

*Song et al. (2017)* propose a 3D oriented task allocation problem. According to different combinations of users, workers and work points, there will be different utility values, and how to maximize it. However, the literature only focuses on the global optimal cost problem, and does not consider the individual situation according to the user and worker. *Li et al. (2020a)* propose a three-class stable matching problem, which combined with the prediction technology in artificial intelligence to provide guidance for online matching, so as to maximize the number of stable matches. *Zheng et al. (2020a)* consider different types of task needs, study how to assign appropriate work points and workers according to task needs, and plan appropriate distribution routes for workers to maximize the total utility. In addition, data security and privacy is also one of the research directions of spatial

crowdsourcing. *Kasım (2022)* study the application of efficient integrated architecture in the privacy and security protection of electronic medical records.

So far, a wide spectrum of incentive mechanisms have been developed for SC systems (*Yang et al., 2016*; *Wang et al., 2019*; *Jin et al., 2018*; *Liu et al., 2020*). But these are mobile-aware crowdsourcing tasks. *El Moussaoui et al. (2023)* study consumers' willingness to use pick-up points. *Tong et al. (2018)* first use incentive mechanism in spatial crowdsourcing. *Tong et al. (2020)* summarize the work of spatial crowdsourcing and mentions the role of incentive mechanism in crowdsourcing. However, the above application of incentive mechanism in spatial crowdsourcing are only limited to the incentive of workers, and it doesn't is no mention the use of incentive mechanism to optimize tasks published by users to achieve the purpose of reducing costs. The research on the security of data by the efficient integrated architecture and the blockchain technology of the Internet of Vehicles is very inspiring for the privacy and security protection of spatial crowdsourced data (*Kasım, 2022*; *Hildebrand et al., 2023*). The work in *Huang, Gao & Gao (2023)* has reference significance for identifying the quality of spatial crowdsourcing service.

The work points mentioned in the literature *Song et al. (2017)* is equivalent to the appropriate pick-up points in the ride-hailing system, but this study are only based on three roles. The fourth role, the destination, is not taken into account. Therefore, it ignore the impact of work point assignment on task execution. In addition, the above literatures only consider the ideal European distance, but the real road network in the real traffic environment is complicated. Users and drivers in the online ride-hailing scene can not shuttle through the map in a straight line. For users, the choice of pick-up point is not a fixed option set in advance, but an open multi-location choice centered on the user's location. Road restrictions information and maximum walking distance limits need to be combined. At present, domestic and foreign researches on online car-hailing mainly focus on the spatial and temporal heat and route planning (*Liao et al., 2022*; *Liao et al., 2023b*; *Liao et al., 2023a*; *Zheng et al., 2020b*), with few studies on pick-up point.

## PROBLEM DESCRIPTION

The subsection "Motivation scenario" shows a motivating example to inspire our research. The problem definitions including the representation of basic definition and problem formalization are described in subsection "Problem definition".

### Motivation scenario

The spatial crowdsourcing task model under the ride-hailing platform is shown in Fig. 4. Users submit travel task requests in real time from their mobile devices, with each task specifying a location, budget, and deadline. After receiving the task request from the user, according to the driver's location, the platform matches according to certain matching rules if the matching result meets the constraints of the current task. The platform generally selects appropriate workers to be assigned to different users from the perspective of improving the probability of completing tasks as much as possible, maximizing the number of tasks completed or minimizing the cost of tasks, *etc.* Finally, both the driver and the user are notified of the current matching results from the platform. The driver needs to arrive at

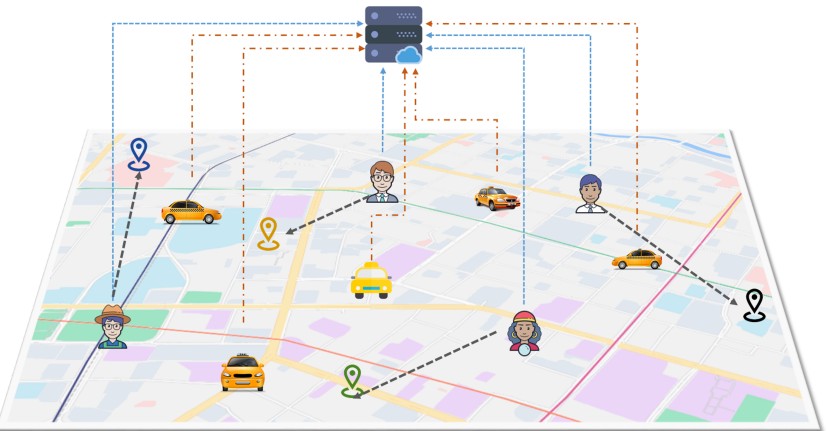

**Figure 4** **Problem crowdsourcing model.**

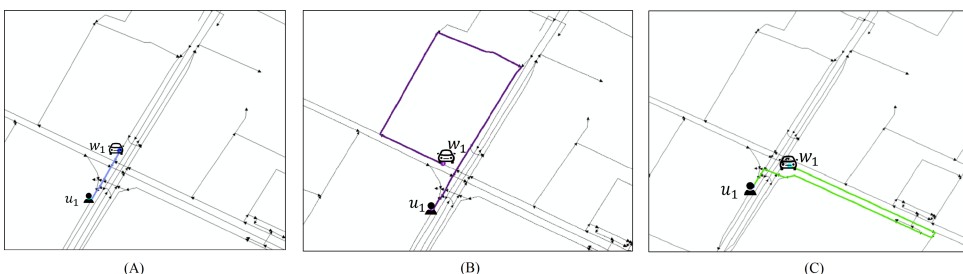

**Figure 5** **Ride-hailing scene.**

the user's location to complete the pick-up service before the deadline of the task request, and then send the user to the destination to complete the execution of the crowdsourcing task.

## Problem scenario

After the platform matches the user with the driver, the driver needs to go to the user's current location. There may be some problems here. For example, Fig. 5 describes a ride-hailing task pick-up scenario.

In Fig. 5, user $u_1$ initiates a car-hailing service at the current location and matches the nearest driver $w_1$ according to the nearest policy, but the current driver $w_1$ is already at the intersection or in the right turn, straight go or left turn lane. When the driver $w_1$ is in the straight go lane, it can quickly receive the user as shown in Fig. 5A. The driver needs to follow the current lane, make a U-turn at the next intersection, and return to the intersection to catch the user, as shown in Figs. 5B and 5C. The appeal case illustrates that fixing the user's location as the pick-up point may greatly increase the execution cost of the task in some cases, so selecting the right pick-up point is an urgent problem to be solved. The goal of this article is to optimize the total cost as an indicator. An incentive mechanism

is proposed to encourage users select the right pick-up point, so as to reduce the overall travel cost of the order and improve the travel efficiency.

## Problem definition

Ride-hailing platform can be regarded as a spatial crowdsourcing platform. Based on current literature and practical application scenarios, this section will introduce some definitions and concepts related to the problem.

**Definition 1 (Worker, $w$).** The spatial crowdsourcing platform includes a set of crowdsourcing workers $W = \{w_1, w_2, \ldots, w_n\}$, worker $w \in W$ are the recipients and performers of the crowdsourcing tasks. They receive and complete the crowdsourcing tasks after registration on the platform. The workers will report their current location $locw_i = (lw_i(x), lw_i(y))$ to the platform, $lw_i(x), lw_i(y)$ respectively indicating the longitude and latitude of the current location of the worker. Each worker is associated with a set of attributes, which are expressed as $w_i = \{locw_i, rw_i, v_i\}$, where $rw_i$ represents the crowdsourcing tasks accepted by the current worker $w_i$ or assigned by the platform, and $v_i$ is the moving speed of the worker on the road section. In the model, the worker accepts the task assigned by the platform. The preference of the worker is not considered in the task assignment process, and the platform assigns tasks to the worker according to the principle of maximizing the global objective function. Each worker can only match one task at a time.

**Definition 2 (Worker, $u$).** The spatial crowdsourcing platform includes a set of users $U = \{u_1, u_2, \ldots, u_m\}$, which are the publishers of spatial crowdsourcing tasks, expressed as $u \in U$. They publish crowdsourced tasks after registration on the platform. When they publish crowdsourced tasks, users will report their current location $locu_j = (lu_j(x), lu_j(y))$ to the platform. Each user is associated with a set of attributes $u_j = \{locu_j, walk_j, ru_j\}$, where $walk_j$ represent the maximum distance that the current user is willing to walk to the surrounding pick-up points, and $ru_j$ represent the crowdsourced tasks published by user $u_j$.

**Definition 3 (Task, $r$).** Spatial crowdsourcing tasks are published by users on the spatial crowdsourcing platform, and each spatial crowdsourcing task usually carries time attributes and geographical location information. The tasks of the online ride-hailing platform are divided into two parts, one is the pre-pick-up task that occurs before the start of the journey, and the other is the journey task after the completion of the pick-up. Represents as $r_k = \{t_k, ori_k, des_k, pri_k, pick_k\}$, where $r_k$ represents the release time of task k, $ori_k, des_k$ represents the original starting point and destination point location of the task k respectively, $pri_k$ isthe order amount of the task, $pick_k$ is the final determined pick-up point location, and the initial value is $locu_j$.

**Definition 4 (Pick-up point, $p$).** For a given task $r_k$, the platform calculates the suitable pick-up point location of the task according to the algorithm combined with the information given. The pick-up point is selected from the road node or interest point, and both user and worker need to go to the selected pick-up point.

**Definition 5 (Task execution cost, $C$).** The completion of each task will have execution cost, and the order cost consists of two parts: distance cost and time cost. Firstly, distance

cost is the sum of the distance cost from the pick-up point to the destination and the distance cost of the driver from the current location to the pick-up point, which is shown as Eq. (1). Secondly, time cost is the sum of the user's waiting time cost and pick-up time cost. As shown in Eq. (2). $dist(O,D)$ represents the road network distance from the origin point $O$ to the destination point $D$, which takes into account the driving direction and turning information of the actual road section, rather than directly using the European distance or Manhattan distance, and can better reflect the true cost of task execution. The total cost of task execution is the sum of distance cost and time cost, which is shown as Eq. (3).

$$C_{r_k}^d \left( w_i, u_j \right) = dist \left( locw_i, ori_k \right) + dist \left( locu_j, ori_k \right) + dist \left( ori_k, des_k \right) \tag{1}$$

$$C_{r_k}^t \left( w_i, u_j \right) = \frac{dist \left( pick_k, des_k \right)}{v_i} \tag{2}$$

$$C_{r_k} \left( w_i, u_j \right) = C_{r_k}^d \left( w_i, u_j \right) + C_{r_k}^t \left( w_i, u_j \right) \tag{3}$$

where $C_{r_k}^d \left( w_i, u_j \right)$ represents the execution distance cost when a crowdsourced task posted by a user is assigned to a worker. $C_{r_k}^t \left( w_i, u_j \right)$ is the corresponding execution time cost. $C_{r_k}(w_i, u_j)$ represents the total execution cost of the task, *kd* and *kt* are distance weight and time weight respectively.

As shown in Eq. (3), if the selected pick-up point is close to both the user and the driver, and the destination can be reached in a convenient way, the task cost will be relatively low. The execution cost of the overall task is calculated by summing the total cost of all tasks, as shown in Eq. (4).

$$C_r = \sum_{r_k \in R} C_{r_k} \left( w_i, u_{j.} \right) \tag{4}$$

**Definition 6 (Incentive mechanism).** In the general spatial crowdsourcing platform, it is always assumed that workers move to the task release point, which will cause detour problems on the real road network. In order to reduce the task execution cost and user waiting time, this article proposes an incentive mechanism to encourage users to walk to the appropriate pick-up point within a certain range, and the reward is used to offset the trip price, which is shown as Eq. (5). Using this as an incentive for users does not require the platform to bear additional costs, but is directly reflected in the reduction of the price of travel orders, which not only reduces the cost of taking the driver, but also reduces the travel consumption for users.

$$R = P(dist(locu_i, des_k)) - P(dist(pcik_k, des_k)) \tag{5}$$

where $P(dist(O,D))$ represents the travel amount under the road network distance from the origin $O$ point to the destination point $D$, which is shown as Eq. (6).

$$P(dist(O,D)) = 10 + dist(O,D) * 3. \tag{6}$$

**Definition 7 (User walking incentive benefit ratio, $Rw$).** In the case that there are multiple pick-up points within a given range, the walking distance of different pick-up points is different, and the execution cost of each pick-up point is also different. Therefore, this article defines the walk incentive benefit ratio to determine the walk benefit of users at each pick-up point, which is shown as Eq. (7). Then the user can choose the pick-up point with high walking incentive income.

$$Rw\left(p_{r_k}^l(u_j)\right) = \frac{dist\left(locu_j, des_k\right) - dist\left(pcik_k, des_k\right)}{dist\left(locu_j, pcik_k\right)} \tag{7}$$

where $p_{r_k}^l(u_j)$ represents the $l$th pick-up point under the user's maximum walking range $walk_j$ in the case of the crowdsourcing task $r_k$ released by user $u_j$, $Rw$ is the user's walking benefit ratio of the pick-up point. It can be seen from the above equation that the greater the distance difference between the pick-up point and the destination point compared with the original starting point and the distance between the user and the pick-up point, the greater the walking benefit of the pick-up point to the user.

**Definition 8 (Driver benefit ratio, $Rd$).** Accordingly, for drivers, the driving costs saved by different driving points are also different, including two driving costs, one is the driving cost of the driver's current location to the driving point, and the other is the driving cost of the driver to the destination from the driving point. Suitable selection of driving points can save the costs of the above two driving stages, so this article defines the driver driving benefit ratio. As shown in Eq. (8), the driving cost and benefit of each driving point are determined.

$$Rd\left(p_{r_k}^l(w_i, u_j)\right) = \frac{dist\left(locw_i, locu_j\right) + dist\left(locu_j, des_k\right)}{dist\left(locw_i, pcik_k\right) + dist\left(pcik_k, des_k\right)} \tag{8}$$

where $p_{r_k}^l(w_i, u_j)$ means that the crowdsourcing task published by the user $u_j$ is assigned to worker $w_i$, and the driving benefit ratio of the driver at the $l$th pick-up point under the maximum walking range $walk_j$ of the user. $Rd$ is the driving benefit ratio of the driver at the pick-up point. It can be seen from the Eq. (8) that the greater the ratio between the original distance of the task and the distance through the pick-up point, the greater the driving benefit of the pick-up point to the driver.

**Definition 9 (Forward rate, $Rf$).** By encouraging the user to walk the appropriate distance to the appropriate pick-up point within the maximum walking distance of the user, the driving distance of the driver can be effectively reduced, and the driving cost of the order execution can be reduced, including the driving cost of the pick-up distance and the driving cost of the drop-off distance in the first stage of the task. However, in the real road network environment, with the increase of the user's maximum walking distance, the number of potential candidate pick-up points around the user increases exponentially. In the case of more pick-up points to choose from, the calculation of each return index of each pick-up point is too large, which cannot meet the needs of real-time interaction scenarios. For this reason, this article defines an indicator of direction rate that includes many factors. As shown in Eq. (9), it can be used to evaluate the score of each pick-up

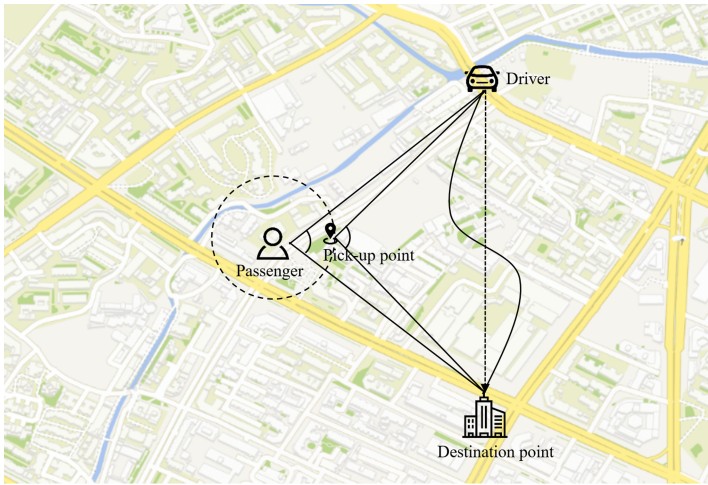

**Figure 6  Forward rate diagram.** Map data ©2023 Tianditu.

point, the diagram is shown in Fig. 6.

$$Rf\left(p_{r_k}^l(w_i, u_j)\right) = \frac{dist(locw_i, des_k)}{dist(pcik_k, des_k)} + \frac{\cos\left(\overrightarrow{pcik_k, des_k}, \overrightarrow{pcik_k, locw_k}\right)}{\cos\left(\overrightarrow{locu_i, des_k}, \overrightarrow{locu_i, locw_k}\right)}. \tag{9}$$

The formula for calculating the forward rate is shown above, which represents the ratio of the driver's direct distance to the destination and the distance from the pick-up point to the destination and the sum of the cosine value from the initial position to the destination and the pick-up point position to the destination. Using the ratio of the two angles instead of the road network distance greatly reduces the calculation cost.

## PICK-UP POINT RECOMMENDATION STRATEGY BASED ON USER INCENTIVE

In order to solve the work point selection problem of spatial crowdsourcing in the scene of online car hailing, this article proposes a pick-up point recommendation strategy based on user incentive (BUIRS). After the platform matches the driver for the user, it gives the user a certain reward to encourage the user to walk to the appropriate pick-up point within a certain range, so as to avoid the detour. The driver also goes to the designated pick-up point from the current location, and then goes to the destination to complete the task after picking up the passenger, which is shown as Fig. 7. BUIRS strategy is divided into the following four steps:

**Step 1 travel characteristics analysis:** The primary purpose of this study is to reduce the overall cost. In general, due to the travel habits of users, most of them prefer to initiate a taxi request at the current location, and hope that the matching driver can go to the fixed point. However, in the peak hours of travel, the crowdsourcing workers on the platform are in short supply, and in this case, the travel efficiency should be improved. Therefore,

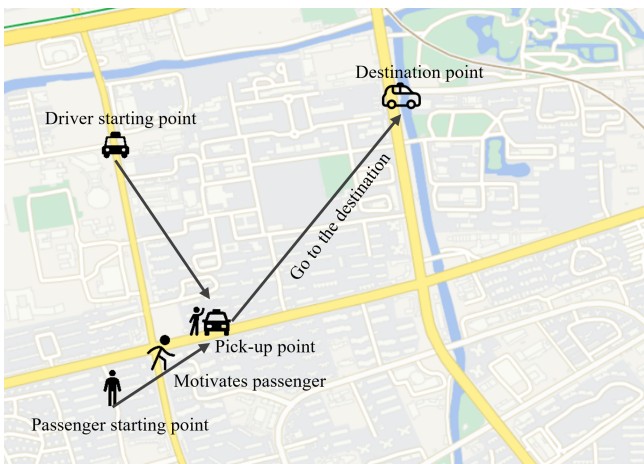

**Figure 7** **The execution diagram of ride-hailing task under user incentive.** Map data©2023 Tianditu.

we select the peak ride-hailing period and corresponding order information as the travel request in this article to verify the effectiveness of the proposed strategy.

**Step 2 driver location algorithm:** Since the information in the data set does not include the location of the driver when receiving orders, this article proposes a simulation to generate the location of the driver when receiving orders according to the order information in the data set.

**Step 3 filter candidate pick-up point:** According to the maximum walking distance and the maximum waiting time of the user, the nearby candidate pick-up point collection meeting the conditions is screened.

**Step 4 recommend pick-up point:** Calculate the indicators of the selected pick-up points in Step 3, and recommend the pick-up points with higher returns to users.

## Travel characteristics analysis

In order to study the order information of peak travel period, it is necessary to analyze the travel characteristic data. This study takes Didi Chuxing Didi Gaia data 2016 Chengdu as an example. This data includes the local order data of Chengdu Second Ring Road in November 2016, which will also be used as the experimental data of this article.

Based on the all-day travel order information on November 1, 2016, heat maps of travel hotspots are generated. In Fig. 8, it can be analyzed from the spatial characteristic distribution of order distribution that the demand for e-hailing data is mainly concentrated in the downtown area, that is, in the third ring road area of Chengdu, especially in the second ring road area, while the demand for travel in other regions is low.

According to the statistics of the order data by hour, the trend of the total order volume per hour is shown in Fig. 9. At 9 am, the demand for online car booking reaches the peak in the morning peak, followed by the second peak demand between 1 pm and 2 pm, and then the evening peak between 1 pm and 2 pm.

The order data are analyzed according to the processing time as shown in Fig. 10. For long-distance travel orders whose processing time is longer than 1 h, the cluster analysis

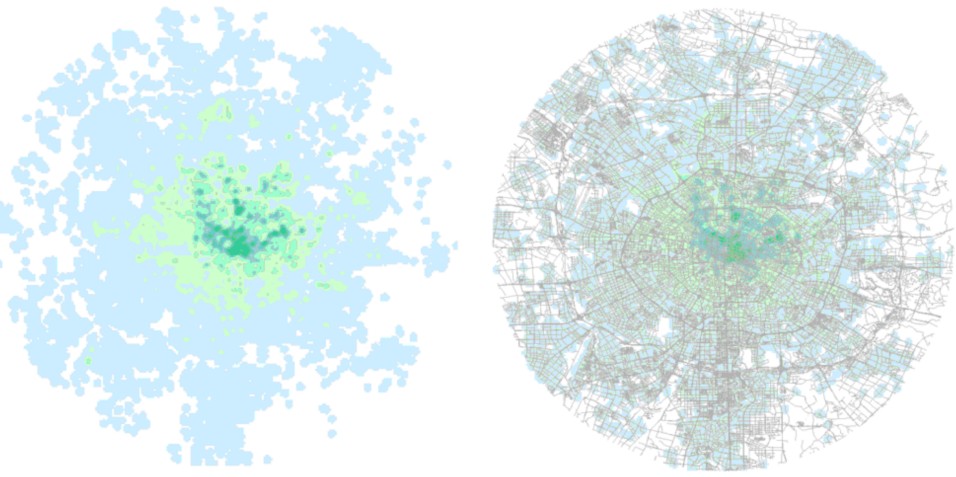

**Figure 8** **Travel demand distribution heat map.** Map data©2023 OpenStreetMap.

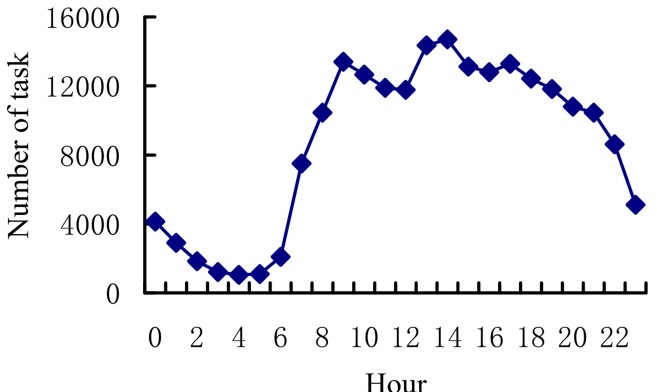

**Figure 9** **Total number of tasks per hour.**

is carried out at five-minute intervals for orders whose processing time is less than 1 h, and the above figure is obtained. As can be seen from the figure, 79% of the orders are processed within 5 to 30 min. The 15-20, 20-25, 25-30 min orders occupy the top three. Based on the above analysis of travel characteristics, this article extracts orders from hot travel areas during peak hours as the input data of this article.

## Driver location algorithm

This section designs a link matching algorithm of anchor points. When an order is assigned to a driver, the location information of the driver needs to be determined. This algorithm is used to generate driver information, ensure that the generated anchor point information

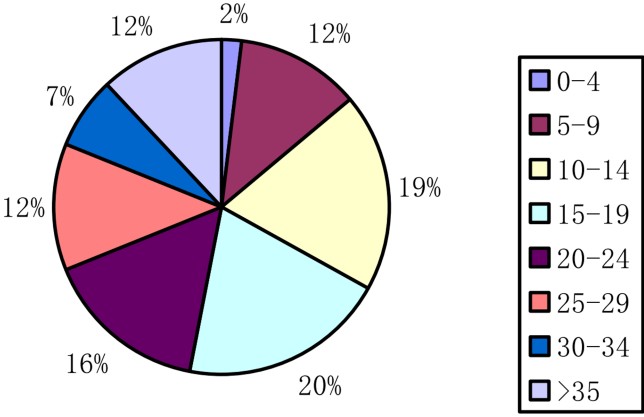

**Figure 10** **Tasks time ratio chart.**

is on the road section, and generate the driving speed according to the information of the road section. The pseudocode of driver location algorithm is shown in Algorithm 1.

---

**Algorithm 1:** Driver location algorithm

**Input:** The task set $R$.

**Output:** The worker set $W$

1  for each task $r_k$ in $R$ do

2      Initial value $w$

3      while $w.lw(x), w.lw(y)$ not satisfies *judgmentconditions*, then

4          $w.lw(x) = ori_k(x) + Random(0.1, 1)$;

5          $w.lw(y) = ori_k(y) + Random(0.1, 1)$;

6      end while

7      $w.v = Random(40, load_{maxspeed})$

8      $W \leftarrow W \cup \{w\}$

9  end for

10  return worker set result $W$

---

The generated driver worker's location and speed information must meet the following restrictions:

Restriction 1: The generated driver location information is within a circle with a radius of 1 km from the center of the user, which is shown as Eq. (10).

$$dist(u_i, w_i) = \sqrt{(u_i(x) - w_i(x))^2 - (u_i(y) - w_i(y))^2} \leq 1 \tag{10}$$

Restriction 2: And it must be satisfied that the anchor point is on the road section. That is the distance from the nearest road side is less than 20 m, which is shown as Eq. (11).

$$dist(locw_i, load) \leq 0.02 \tag{11}$$

The initial speed of the driver is set to be less than the maximum speed of the road section, which is shown as Eq. (12).

$$40 \leq w_{iv} \leq load_{maxspeed}. \tag{12}$$

## Filter candidate pick-up point

When a user launches a crowdsourcing task, there are two location information, the user origin point $O$ and the destination location $D$, in addition, the algorithm is designed based on the user incentive mechanism to recommend the user to the appropriate pick-up point, the user's maximum walking distance is $walk_j$, and the cost of motivating the user to walk is $i$. When the user initiates an order, the platform finds a suitable collection of pick-up $P_k = \{p_1, p_2, \ldots, p_m\}$ points according to the user's destination and the current user's location, and calculates the order cost.

The original task execution cost is obtained from Eq. (3). After selecting the optimal pick-up point, the distance cost of the order becomes the sum of the distance cost from the pick-up point to the destination and the distance cost from the current location to the pick-up point. Secondly, the time cost becomes the sum of the time from the user's current location to the boarding point and the maximum time from the driver to the boarding point and the time from the user to the destination after the boarding.

According to the user's maximum walking distance and maximum waiting time, the algorithm first selects the candidate pick-up point collection that meets the conditions nearby. The pseudocode of filter candidate pick-up point algorithm is shown in Algorithm 2.

---

**Algorithm 2:** Filtrate candidate pick-up point algorithm

**Input:** The user $u_j$, task $r_k$, road network $N$

**Output:** The candidate pick-up point set $P_k$ of task $r_k$

1   Initial value $P_k \leftarrow \emptyset$

2   for each point $p \in N$

3     if $p$ satisfies *judgmentconditions*, then

4       $P_k \leftarrow P_k \cup \{p\}$

5   end for

6   return candidate pick-up point set result $P_k$

---

The candidate pick-up point need to meet the following restrictions:

Restriction 3: The selection of the pick-up point needs to meet the distance from the user is less than the user's maximum walking distance, which is shown as Eq. (13).

$$dist\left(u_i, p_{i,j}\right) = \sqrt{\left(u_i(x) - p_{i,j}(x)\right)^2 - \left(u_i(y) - p_{i,j}(y)\right)^2} \leq u_{i,w} \tag{13}$$

Restriction 4: Time limit, the selected pick-up point should ensure that the driver can arrive within the maximum waiting time of the user, which is shown as Eq. (14).

$$\frac{dist\left(w_k, p_{i,j}\right)}{v_k} \leq \left(t_i + wt_i - p\right) \tag{14}$$

## Recommend pick-up point
### *Pick-up point recommendation strategy based on user incentive-greedy (BUIRS-G)*

After selecting qualified candidate pick-up points, this article proposes BUIRS algorithm BUIRS-G based on greedy strategy, which aims at cost optimization. The algorithm calculates the task execution cost, user incentive value and other parameters of each pick-up point one by one according to the corresponding formula. The more profitable pick-up points are recommended to the user, who can then choose the appropriate pick-up points according to the specific situation. The pseudocode of BUIRS-G algorithm is shown in Algorithm 3.

---

**Algorithm 3:** Recommendation strategy based on user incentive-greedy (BUIRS-G)

---

**Input:** The user $u_j$, worker $w_i$ task $r_k$, candidate pick-up point set $P_k$, road network $N$

**Output:** The recommend pick-up point set $P_k$ of task $r_k$

1    for each point $p \in P_k$
2        $C_p \leftarrow C_{r_k} w_i, u_j, p$
3        $r_p \leftarrow reward_p u_j, r_k, p$
4        $Income\_w_p \leftarrow Rw\left(p^p_{r_k}(u_j)\right)$
5        $Income\_d_p \leftarrow Rw\left(p^p_{r_k}(w_i, u_j)\right)$
6    end for
7    for each point $p \in P_k$
8        if $p$ not satisfies *judgmentconditions*, then
9          $P_k \leftarrow P_k \backslash \{p\}$
10   end for
11   return the recommend pick-up point set result $P_k$

---

Algorithm 3 firstly calculates the task execution cost of each candidate pick-up point (line 2) according to Eq. (3), and then calculates the user incentive value of each pick-up point (line 3) and the user walking benefit ratio to driver driving benefit ratio of each pick-up point (line 4–5) according to the task cost and Eq. (5). Then filter out the pick-up points where the incentive value does not meet the user's expectations (line 8). Finally, the pick-up point with high revenue value is recommended to the user, and then the user can choose the right pick-up point according to the specific situation. The incentive value of the candidate pick-up point should be greater than the user's minimum expected incentive value, which is shown as Eq. (15).

$$reward_p\left(u_j, r_k, p\right) = \left(dist\left(O, D\right) - dist\left(u_{pick}, O\right)\right) * c \le u_{i,w} \tag{15}$$

### *Pick-up point recommendation strategy based on user incentive-forward rate (BUIRS-F)*

In Algorithm 3, when the maximum walking distance set by the user is relatively large, there will be a large number of potential candidate pick-up points, and it will take a long time to

calculate the task execution cost and benefit of each pick-up point one by one. In order to meet the real-time computing requirements, the forward rate index mentioned in Related Work is replaced by the execution cost and benefit index for calculation. BUIRS algorithm based on forward rate BUIRS-F is proposed. The pseudocode of BUIRS-F algorithm is shown in Algorithm 4.

---

**Algorithm 4:** Recommendation strategy based on user incentive-forward rate(BUIRS-G)

---

**Input:** The user $u_j$, worker $w_i$ task $r_k$, candidate pick-up point set$P_k$, road network $N$

**Output:** The recommend pick-up point set$P_k$ of task $r_k$

1   for each point $p \in P_k$
2       $Rf_p \leftarrow R_f \left( p_{r_k}^p \left( w_i, u_j \right) \right)$
3   end for
4   $P_k \leftarrow \varnothing, sort(Rf), Rf = front(Rf)$
5   for each point $Rf_p \in Rf$
6       if $p$ satisfies *judgmentconditions*, then
7           $P_k \leftarrow P_k \cup \{p\}$
8   end for
9   return the recommend pick-up point set result $P_k$

---

In Algorithm 4 firstly calculates the forward rate of each candidate pick-up point (line 2) according to Eq. (9), clears the set of candidate pick-up points after the end, sorts each pick-up point in descending order according to the forward rate, and intercepts the set of pick-up points ranked in the first part (line 4). Then only the user incentive value of the intercepted pick-up point is calculated (lines 5–6). The reduction of the driver's pick-up cost is the sum of the two distance distances with the pick-up point as the transit point. The user benefits and driver benefits of each pick-up point are calculated (lines 4–5), and then the pick-up points whose incentive values meet the user's expectations are added to the recommendation set (line 8).

## SIMULATION EXPERIMENT ANALYSIS

This section mainly describes the data set used in the experiment, and tests the influence of the forward rate index mentioned in this article on the selection of pick-up points in the data set. Then the algorithm proposed in this article is applied to solve the problem of the selection of the pick-up point, and finally the experiment is carried out to measure the influence of different parameters on the results

### Simulation environment and data set

The data set used in this article is the data set of Didi Dache in Chengdu, China in November 2016. The experiment uses Python language. The environment of simulation experiment is 64-bit Windows 10 system, memory (RAM) is 16GB, and the processor is Intel(R) Core(TM) i5-9400F CPU @ 2.90 GHz. In order to test the algorithm proposed in this article, part of the data set of Didi Dache in Chengdu, China, on November 1, 2016, was

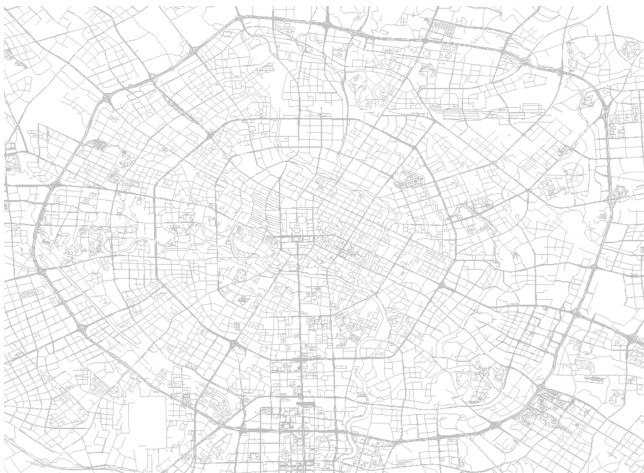

**Figure 11**  **Chengdu road network data display map.** Map data ©2023 OpenStreetMap.

extracted according to the spatial and temporal characteristics of peak hours, including 4,824 order data, including the start and end time of orders and the latitude and longitude of getting on and off the bus. The coordinate system was GCJ-02. The road network data set is the road network vector data of Chengdu, Sichuan Province.

Based on the order information during peak hours, this study intercepts the road network information within the 4th ring road of Chengdu, including 46,661 interchanges and 131,366 road side information. The vector road network information includes road side node information, road grade, road name, one-way and dual-way traffic restriction information and maximum speed limit information. The road network information is shown in Fig. 11.

For the Didi real data set, we use the pick-up and drop-off locations in the order to initialize the location of the user and destination in the pick-up point recommendation question. We process each order in turn according to the start time of processing in the real data set. The waiting time for each order is randomly selected in [4,10]. The location of the driver is randomly generated on the road within a circle with a radius of 1 km from the center of the user; The speed v of each driver is set in the range of [40, maxspeed] according to the road speed limit. Set the maximum movement distance for each user to 400 m.

## Performance analysis

Five evaluation indicators are introduced in this section, namely time cost, average response time, energy consumption cost, the number of successful matches and matching success rate.

(1) Analysis of various indicators of pick-up point

This section first shows the selection of a pick-up point for an task in the experiment. The recommended pick-up point of the user $u_1$ is shown in Fig. 12. The star is the location of the user, the gray is each candidate pick-up point, and the blue is the pick-up point

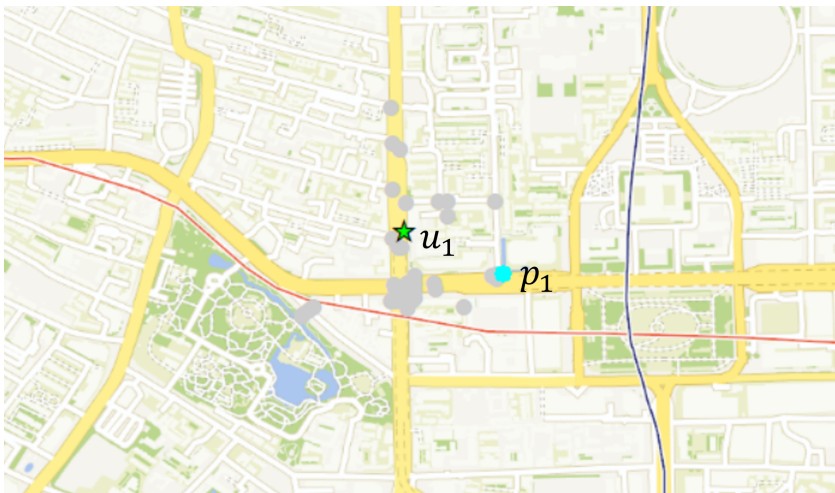

**Figure 12**  $u_1$'s pick-up point recommendation result.  Map data ©2023 Tianditu.

recommended by the algorithm. Figure 13 shows the indicators of the top 12 pick-up points in task execution cost.

As can be seen from Fig. 13, pick-up point $p_1$ is larger in the walking distance of users, but the difference is not very obvious compared with other candidate pick-up points. Pick-up point $p_1$'s incentive value is in the middle position in terms of the benefit ratio of walking distance, but it is obviously better than other candidate points pick-up distance. In the fourth indicator of drop-off distance, the pick-up point $p_1$ also occupies a large advantage, and the algorithm selects the point $p_1$ in line with expectations.

(2) Analysis of the impact of maximum walking distance on total cost

Figure 14 shows the cost comparison diagram between the user's original travel distance and the shortest travel distance within each walking distance. Among them, the yellow line segment is the original travel distance, the blue bar is the total travel distance of each optimal pick-up point under different user maximum walking distance restrictions, and the orange line is the actual user walking distance. It can be seen that the total cost of the task is gradually decreasing while the maximum walking distance limit is increasing. This is because when the maximum walking distance of users continues to increase, the number of available pick-up points increases, and users can have better pick-up point selection schemes. In the process of increasing the range from 50 m to 200 m, although the number of candidate pick-up points increases, the new pick-up points cannot further optimize the execution cost, so the choice of pick-up points remains unchanged. The execution cost of the order has not changed either, but as the maximum walking distance of the user has been further increased, better pick-up points have emerged, and the execution cost of the order has decreased.

(3) Analysis of indicators of forward rate

When the maximum walking distance of the user increases, the corresponding number of pick-up points increases exponentially. Figure 15 shows the number of alternative

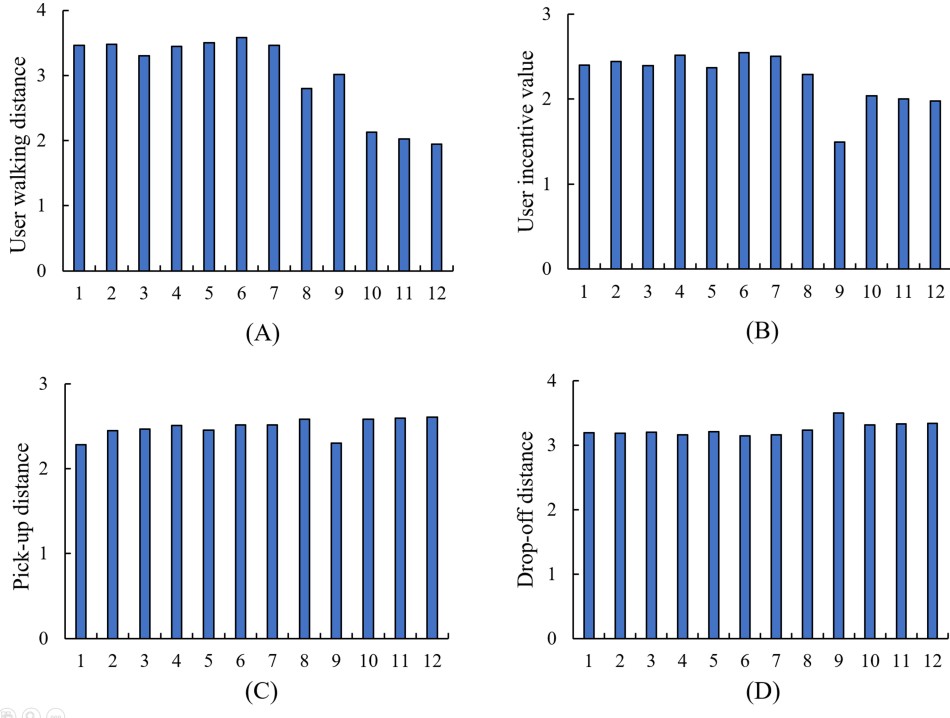

Figure 13    Comparison of pick up point indexes.

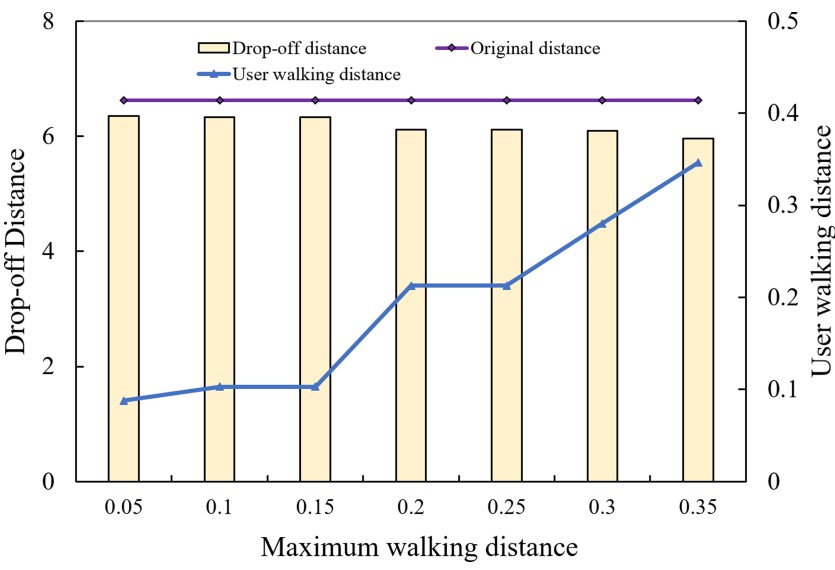

Figure 14    Walking distance and task cost.

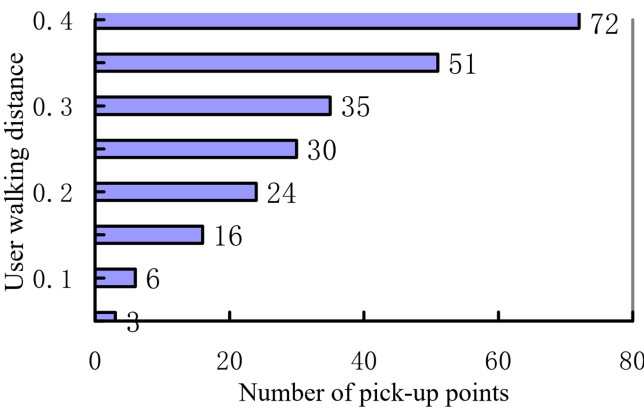

**Figure 15  Number of pick-up points and walking distance.**

pick-up points of the user under different maximum walking distance restrictions. When the maximum walking distance is increased from 50 to 400 m, the number of candidate pick-up points increases from 0 to 72. When the limit is further expanded, it can be seen that the number of candidate pick-up points increases geometrically.

Because the actual distance calculation in this article does not take the simple Euclidean distance as the distance between two points, but the actual road network distance as the standard. The distance calculation involves the search for the shortest path of the graph and includes the traffic restriction information of each section. The search for the optimal option is not the best choice in calculating the order execution assembly corresponding to each pick-up point under the platform with high real-time requirements, which consumes a lot of computing resources on the platform. In this section, the forward rate index proposed in the second section is experimentally verified.

For the user $u_1$, under the limitation of the maximum walking distance of 400 m, the relationship between the sequence and the direction rate of each point is obtained by ordering all the candidate pick-up points according to the task distance cost, as shown in Fig. 16. As can be seen from the figure, with the increase of total distance cost, the forward rate of pick-up points decreases. Although a few pick-up points in the middle have higher forward rates, the overall downward trend will not be affected. The objective of this article is to find relatively optimal pick-up points under the condition of optimization of computational load, and if only the optimal pick-up points are selected, the real-time response cannot be satisfied under the condition of a large number of user requests. The candidate pick-up points whose forward rate is lower than a certain threshold can be discarded in subsequent calculation, which greatly reduces the loss of computing resources.

Figure 17 shows the trend chart of task execution cost and forward rate for the first 12 alternative pick-up points of user 1. As can be seen from the figure, the pick-up points with low distance cost have a higher turn-along rate. Therefore, the turn-along rate index proposed in this article can represent the pick-up points' turn-along rate to a certain extent. When recommending and screening pick-up points to users, the platform can first determine the top several alternative pick-up points according to the pick-up rate, and

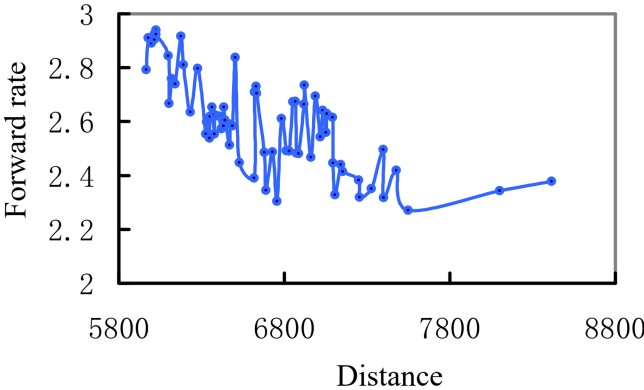

**Figure 16   Forward rate and task cost of pick-up point.**

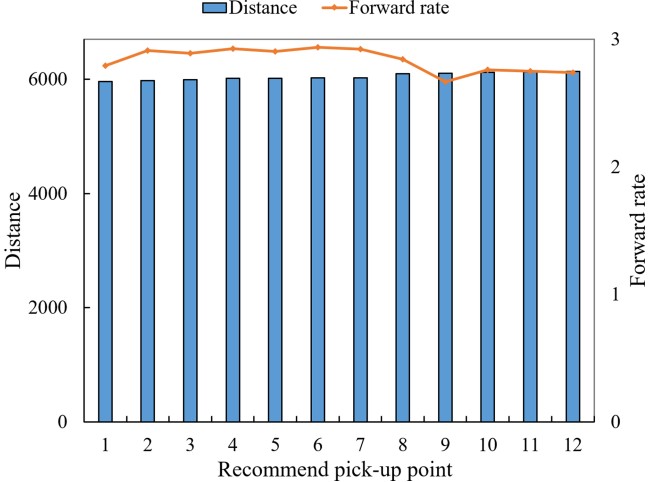

**Figure 17   The forward rate of the top 12 candidate points.**

then continue to calculate other indicators to determine the optimal pick-up points. Save computing resources.

## Comparative analysis

In this section, a performance comparison of methods used to solve the pick-up point selection problem is described. Pick-up point recommendation strategy based on user incentive BUIRS-G greedy algorithm and BUIRS-F forward rate algorithm are compared with *Song et al. (2017)* Three-dimensional matching recommendation strategy (TSMRS) and random recommendation strategy (RRS) in the indicators of task execution cost and running time.

Specifically, in each round of algorithms, the platform selects the best pick-up point for users in each order according to the two recommendation strategies mentioned in this article. In the experiment in this article, BUIRS-G defaults to the pick-up point with the

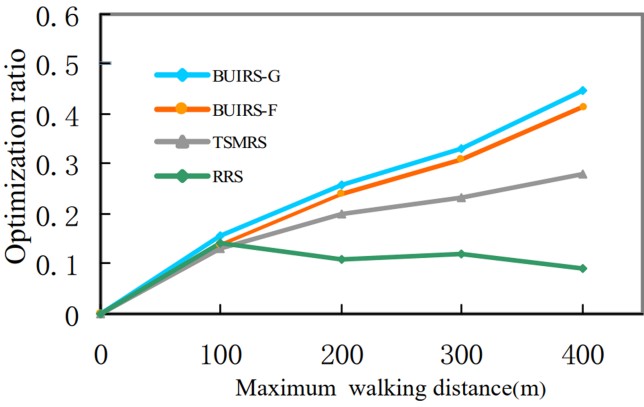

**Figure 18** Task cost optimization ratio.

smallest task execution cost as the best pick-up point, and BUIRS-F defaults to the pick-up point with the highest forward rate as the best pick-up point. In contrast, in TSMRS, users, workers and pick-up point are considered to minimize the moving distance between users and workers as the solution goal to obtain the optimal pick-up point. In each round, the algorithm pays attention to the shortest move in the first stage, but ignores the trip information in the second stage and does not consider the overall income of the selected pick-up point, which may lead to high driving cost in the second stage.

Through the selected order data of Didi Travel during peak hours, the influence of the optimal pick-up point selected under different maximum walking distance limits of 0.1 km, 0.2 km, 0.3 km and 0.4 km on the total order cost is set, as shown in the figure.

Figure 18 shows the task optimization ratio of each method under the maximum walking limit of different users. The calculation method is the ratio of the task execution cost under the optimal pick-up point selected by each method to the task execution cost directly through the user's location. Figure 19 shows the time cost under each method. BUIRS-G proposed in this article is better than the other three methods in terms of cost optimization, followed by BUIRS-F, TSMRS method is not as good as the first two methods in terms of overall optimization because it ignores the task cost in the second stage.

In terms of time cost, BUIRS-G calculates each indicator of candidate pick-up points one by one, so the speed is always slow, which is more obvious when the maximum walking limit increases. Because the number of pick-up points increases significantly, BUIRS-F method only needs to calculate the forward rate, and uses the pick-up points with a higher forward rate to calculate the user incentive value. Therefore, the time overhead is small. TSMRS method needs to calculate the task cost of the first stage one by one, and the running time is smaller than BUIRS-G method. RRS is a random selection of pick-up points, each selection has randomness, and the optimization cost is low. In terms of time overhead, there is no calculation for random selection, so the overhead is the least.

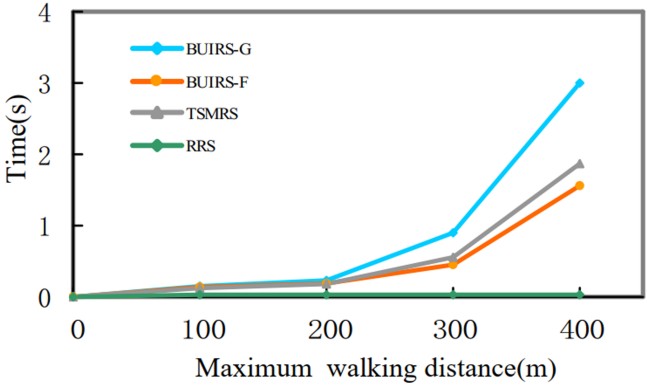

**Figure 19** Time cost.

## CONCLUSIONS

This article presents a new problem in the practical application of spatial crowdsourcing, that is, a crowdsourcing task pick-up point recommendation problem with four dimensions. Compared with the way that workers rush to the user's location in general crowdsourcing tasks, this study proposes the concept of user incentive. Based on the perspective of optimizing task execution cost, users are encouraged to go to the nearby pick-up point. Based on this, this article proposes a pick-up point recommendation strategy to solve this problem, and proposes BUIRS-G algorithm based on greedy strategy and BUIRS-F algorithm based on forward rate strategy. Finally, experiments are carried out to prove the effectiveness and timeliness of the proposed algorithm by comparing various algorithms, which provides decision support for the problem of pick-up point recommendation.

Experiments in this study show that the proposed pick-up point recommendation algorithm can reduce the overall travel cost by about 20% on average. Furthermore, the proposed algorithm can complete the calculation faster than others while ensuring the effectiveness of the recommendation results. In addition, the result of this strategy will reduce unnecessary detour distance and energy consumption of drivers while performing tasks, improving the efficiency of crowdsourcing task execution. At the same time, it also reduces the order amount and improves user satisfaction. That is, this study will optimize the recommendation of the location of the pick-up point in the online ride-hailing service by combining the theory of spatial crowdsourcing. To achieve more accurate, practical and more satisfying pick-up point location recommendation, so as to provide more convenient services for urban travel, reduce traffic congestion and energy waste and other problems.

However, the existing pick-up-point recommendation strategy is not very perfect, and some factors are not considered enough, such as the congestion problem that may be caused by assigning the same pick-up point to multiple tasks at similar locations. More factors will be considered in the follow-up pick-up point recommendation research. In

addition, the issue of the impact of the pick-up point on user privacy also needs special attention, which is one of the directions of future research.

### Funding

This work was supported by the National Natural Science Foundation of China (No. 61902069), the Natural Science Foundation of Fujian Province of China (2021J011068), and the Special Fund for Scientific Research of Fujian Provincial Department of Finance in 2022. The funders had no role in study design, data collection and analysis, decision to publish, or preparation of the manuscript.

### Grant Disclosures

The following grant information was disclosed by the authors:
National Natural Science Foundation of China: No. 61902069.
Natural Science Foundation of Fujian Province of China: 2021J011068.
Special Fund for Scientific Research of Fujian Provincial Department of Finance.

### Competing Interests

The authors declare there are no competing interests.

### Author Contributions

- Jing Zhang conceived and designed the experiments, analyzed the data, authored or reviewed drafts of the article, and approved the final draft.
- Biao Li conceived and designed the experiments, performed the experiments, analyzed the data, performed the computation work, prepared figures and/or tables, authored or reviewed drafts of the article, and approved the final draft.
- Xiucai Ye performed the experiments, analyzed the data, performed the computation work, prepared figures and/or tables, authored or reviewed drafts of the article, and approved the final draft.
- Yi Chen performed the experiments, analyzed the data, performed the computation work, prepared figures and/or tables, authored or reviewed drafts of the article, and approved the final draft.

### Data Availability

 The raw data and code are available in the Supplemental File.

### Supplemental Information

Supplemental information for this article can be found online at http://dx.doi.org/10.7717/peerj-cs.1692#supplemental-information.

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
