# Peer review of "Pick-up point recommendation strategy based on user incentive mechanism"

_PeerJ Computer Science, doi:10.7717/peerj-cs.1692_

## Round 0.1 · original submission · Major Revisions

The authors proposed a reasonable method of crowdsourcing. However, there are some issues in writing. Some details are not described clearly. Please revise the work accordingly. It will be evaluated again.

**Language Note:** PeerJ staff have identified that the English language needs to be improved. When you prepare your next revision, please either (i) have a colleague who is proficient in English and familiar with the subject matter review your manuscript, or (ii) contact a professional editing service to review your manuscript. PeerJ can provide language editing services - you can contact us at copyediting@peerj.com for pricing (be sure to provide your manuscript number and title). – PeerJ Staff

Reviewer 1 ·

Basic reporting

The authors proposed a method of crowdsourcing. They presented a design in 4d compared to other methods. They shared their test results with different algorithms of this design. The work was generally well prepared. But it could get better with the following additions.

* A sentence about hypothesis and motivation should be added to the introduction.

* The success achieved should be mentioned in the conclusion section. In addition, it should be stated what kind of benefits the result will be.

* The state of art of the work should be developed with the following resources. Data security of data should also be included.

Seng, K. P., Ang, L. M., Ngharamike, E., & Peter, E. (2023). Ridesharing and Crowdsourcing for Smart Cities: Technologies, Paradigms and Use Cases. IEEE Access, 11, 18038-18081.

El Moussaoui, A. E., Benbba, B., Jaegler, A., El Moussaoui, T., El Andaloussi, Z., & Chakir, L. (2023). Consumer perceptions of online shopping and willingness to use pick-up points: A case study of Morocco. Sustainability, 15(9), 7405.

Xie, C., Wen, X., Pang, H., & Zhang, B. (2023). Application of graph auto-encoders based on regularization in recommendation algorithms. PeerJ Computer Science, 9, e1335.

November, O. (2022). An Efficient Ensemble Architecture for Privacy and Security of Electronic Medical Records. int. Arab J. Inf.
Technol., 19(2).

Hildebrand, B., Tabassum, S., Konatham, B., Amsaad, F., Baza, M., Salman, T., & Razaque, A. (2023). A comprehensive review on blockchains for Internet of Vehicles: Challenges and directions. Computer Science Review, 48, 100547.

Huang, L., Gao, B., & Gao, M. (2023). Acceptance and Use of Omni-Channel Retail Services (Segment Analysis). In Value Realization in the Phygital Reality Market: Consumption and Service Under Conflation of the Physical, Digital, and Virtual Worlds (pp. 125-150). Singapore: Springer Nature Singapore.

** These references are offered as suggestions. Additional resources can be added as well.

Experimental design

.

Validity of the findings

.

Additional comments

.

Cite this review as

Reviewer 2 ·

Basic reporting

In this article, the authors investigated the crowdsourcing task pick-up point recommendation problem by designing an incentive mechanism to encourage users to walk to the nearby pick-up point so that the execution efficiency of crowdsourcing tasks can be improved. According to the comparative analysis reported in Figure 18, the efficiency optimization ratio is significantly upgraded as users’ tolerable walking distance increases. This result is reasonable.

Experimental design

However, the distance cost is the sum of the distance from the pick-up point to the destination and the distance of the driver from his/her current location to the pick-up point (Eq. (1)), ignoring the distance cost from the user’s original location to the pick-up point. The authors should explain in more detail whether the defined task execution cost can meet the practical situations.

Validity of the findings

My other comments are listed below:
1. Page 6, line 224: “workw_i” is a typo.
2. Page 7, line 232: “locu_i” should be “locu_j”.
3. Page 7, Eq. (1): “locw” and “ori” should be italic.
4. Page 7, Eq. (3): “kd” is confusing. Does it mean k times d?
5. Page 7, Eq. (5): The last right-parenthesis is missing.
6. Page 8, Eq. (9): R_f should be Rf, fitting the notation in line 295.
7. Page 11, Algorithm 1: Step 8 should be W <-- W U {w}.
8. Page 12, Algorithm 2: Step 3 should be P <-- P U {p}
9. Page 17, Figure 18: The authors should also show the values of optimization ratio for the maximum walking distance being zero. This case can be seen as no pick-up points involved.

Additional comments

This paper can be reconsidered for publication after minor revision if my comments are fully addressed.

Cite this review as

---

## Round 0.2 · accepted · Accept

Congrats to the authors. The current version could be accepted.

Reviewer 1 ·

Basic reporting

Necessary corrections and adjustments have been made. It is appropriate to publish it in this form.

Experimental design

Necessary corrections and adjustments have been made. It is appropriate to publish it in this form.

Validity of the findings

Necessary corrections and adjustments have been made. It is appropriate to publish it in this form.

Additional comments

Necessary corrections and adjustments have been made. It is appropriate to publish it in this form.

Cite this review as

Reviewer 2 ·

Basic reporting

In this article, the authors investigated the crowdsourcing task pick-up point recommendation problem by designing an incentive mechanism to encourage users to walk to the nearby pick-up point so that the execution efficiency of crowdsourcing tasks can be improved.

Experimental design

The proposed algorithm is applied to solve the problem of the selection of the pick-up point, and a simulation experiment is carried out regarding the data set of Didi Dache in Chengdu, China in Nov. 2016 to evaluate the influence of different parameters on the results.

Validity of the findings

According to the comparative analysis reported in Figure 18, the efficiency optimization ratio is significantly upgraded as users’ tolerable walking distance increases.

Additional comments

The authors made a satisfactory revision by fully addressing my comments. I recommend this paper to be accepted for publication.

Cite this review as